# The Interaction and Its Evolution of the Urban Agricultural Multifunctionality and Carbon Effects in Guangzhou, China

Zuxuan Song [1] and Ren Yang [1,2,*]

1   School of Geography and Planning, Sun Yat-sen University, Guangzhou 510006, China
2   Land Research Center, Sun Yat-sen University, Guangzhou 510275, China
*   Correspondence: yangren666@mail.sysu.edu.cn

**Abstract:** The dual identity of carbon sources and carbon sinks makes agriculture the focus of carbon neutralization-related research. Compared with traditional rural agriculture and urban industrial production, urban agriculture has its own particularities. It is of positive practical significance to explore the interaction and its evolution process between urban agricultural multifunctionality and carbon effects in seeking solutions to alleviate carbon pressure. Based on the changes in agricultural carbon emissions and carbon sequestration in Guangzhou from 2002 to 2020, we used the Granger causality analysis method to investigate the interaction between urban agricultural multifunctionality and carbon effects and then used the grey association model to analyse the evolution process of associative degrees between the two and divide the agricultural development stages. Finally, according to the practicalities of Guangzhou, we analyzed carbon effects generated in the multifunctional transformation of urban agriculture and put forward corresponding policy suggestions on how to solve the problem of excessive carbon dioxide emissions through agriculture in metropolitan areas. The results show that from 2002 to 2020 in Guangzhou, urban agricultural production decreased, the economic and social function increased, and the ecological function climbed and then declined. The carbon sequestration of urban agriculture in Guangzhou was approximately four times more than the carbon emissions. Carbon emissions experienced a process of first decreasing, then increasing, then remaining constant, and finally decreasing, while carbon sequestration first decreased and then increased. Second, the carbon emissions of urban agriculture in Guangzhou have a causal relationship with the production, social, and ecological functions. Carbon emissions are the Granger cause of the economic function but not the opposite. The carbon sequestration of urban agriculture in Guangzhou has a causal relationship with production and economic functions. Carbon sequestration is the Granger cause of the ecological function but not the opposite. There is no Granger causal relationship between carbon sequestration and the social function. Third, from 2002 to 2020, the interactive development process of urban agricultural multifunctionality and carbon effects in Guangzhou can be divided into three stages: production function oriented (2002–2006), economic and social function enhanced and production function weakened (2007–2015) and the economic and social function exceeded the production function (2016–2020). Fourth, the multifunctional transformation of urban agriculture has brought about carbon effects of reducing emissions and increasing sequestration. There is a long time lag between multifunctional transformation and carbon effects of urban agriculture.

**Keywords:** carbon effects; urban agricultural multifunctionality; Granger causality analysis; grey association model; Guangzhou



## 1. Introduction

Climate change is a major risk faced by mankind, and the emission of carbon dioxide (CO$_2$) is considered to be one of the main causes of Earth's climate change [1]. The excessive emission of CO$_2$ has had a serious negative impact on the global ecosystem and socioeconomic system, such as sea level rise, damage to biodiversity and reduction of agricultural productivity [2,3]. To cope with the deteriorating environmental quality, the

world has reached a consensus to formulate policies to reduce carbon emissions [4], such as the Kyoto Protocol in 1997 and the Paris Agreement in 2016 [5]. In 2016, China's carbon dioxide emissions accounted for approximately one-third of the global total [6]. In view of China's responsibilities in the agreements of global carbon emission reduction and the proposal of the "double carbon" (achieving carbon peak in 2030 and carbon neutralization in 2060) strategy, understanding the status quo of carbon emissions and carbon sequestration in China is of great significance to the formulation and implementation of carbon emission reduction policies. With China's rapid urbanization, cities have become the main areas of energy consumption, and the proportion of carbon emissions in cities has reached 85% [7]. Therefore, cities occupy a core leading position in seeking solutions and mitigation strategies for excessive $CO_2$ emissions [8,9].

In the carbon cycle within the ecosystem, there is a rapid exchange of $CO_2$ among the three main compartments of atmosphere, phytosphere and pedosphere without disturbance to the system balance so that the ratio of carbon content among them is relatively stable [10]. In human agricultural activities, ploughing destroys the soil organic carbon. Mechanization has made agricultural production increasingly dependent on fossil fuels [6]. Agricultural inputs such as pesticides and fertilizers, the processing and distribution of agricultural products and the disposal and utilization of agricultural waste lead to energy consumption and loss in agriculture to varying degrees, generating large amounts of fossil fuel carbon. Agricultural greenhouse gas emissions account for approximately 17% of China's total emissions [11]. However, at the same time, the crop production system is also a large carbon pool. $CO_2$ increases photosynthesis and the intrinsic water use efficiency of leaves, and these direct reactions transfer atmospheric carbon to terrestrial ecosystems [12]. Therefore, the planting production system plays a very important role in maintaining agricultural soil carbon sinks, improving the level of soil organic carbon and ensuring food security [13]. The dual role of carbon sources and carbon sinks makes agriculture the focus of carbon neutralization related research [14].

Urban agriculture has accompanied the development of modern cities. To make full use of natural resources such as cultivated land, mountains and water surfaces [15] while relying on superior human and industrial resources, urban agriculture closely serves cities, integrates into the urban economic system and finally becomes an organic part of the urban economy, society and ecosystem. Improved transportation and technology infrastructure has made urban agriculture more competitive [16]. To meet the demands of cities for products and services, agriculture has turned from traditional planting to multifunctionality [17,18]. This restructuring transcends the traditional production mode based on pure commodity production, turning to a new production system that provides consumers with other goods and services [19]. Multifunctional agriculture is essentially a post-productivist agricultural model [20], which can support environmental sustainability and rural development [21]. Urban agriculture combines knowledge from the traditional agricultural sector with a range of new skills, technologies, tools and strategies [22] and has a variety of economic and non-economic benefits. Economic benefits include local employment and local agribusiness growth [23]. Non-economic benefits include recreational and cultural heritage, community solidarity, quality of life, educational and medical opportunities and tourism development [24]. Agricultural multifunctionality is a part of agricultural land output [25]. FAO (1999) attempted to formulate an overall planning framework that includes food security, environment, economic and sociocultural functions [26]. Compared with traditional agricultural production, urban agriculture has stronger scientific and technological advantages and more abundant sales channels, so it has higher production efficiency and economic benefits. The transformation of production modes, the application of ecological economy and the deepening of the low-carbon concept have made urban agriculture generate carbon effects different from traditional agriculture. On the other hand, urban industrial production only pursues economic benefits, burns a large amount of fossil fuels and emits carbon dioxide, which is a socio-economic system completely controlled by human activities. Therefore, it is difficult to make contributions to carbon sequestration. In

contrast, urban agriculture is a regional system where human beings and nature coexist in harmony and supplement each other. While pursuing production benefits through science and technology, it also takes the positive impact of crops on the environment into account and therefore promotes the carbon balance of the ecosystem. An important academic topic in the scientific study of modern agriculture is the interaction mechanism between urban agricultural multifunctionality and carbon effects, which has positive practical significance in seeking solutions to alleviate carbon pressure.

The agricultural carbon footprint in China mainly comes from farming in the northwest, the agricultural machinery used in farming in north China, and the fertilizers used in the other four areas. Carbon sinks are mainly sugarcane in south China, rice in the middle and lower reaches of the Yangtze River and corn in northeast China [13]. In terms of the changes in agricultural carbon effects, the urban greening policies designed to mitigate the carbon footprint caused by urban construction land expansion, vegetation restoration in areas of rural emigration and the slight decline in arable land brought a large but transient carbon sink. Rapid urbanization and carbon neutralization are not mutually exclusive. A significant reduction in $CO_2$ emissions from fossil fuel burning is key to achieving carbon neutralization [27]. In terms of the influencing factors of agricultural carbon emissions, urbanization, economic growth, financial capacity and energy efficiency have a greater impact on provinces with high carbon emissions [28]. There is a weak and unstable decoupling relationship between agricultural carbon emissions and output value. The urbanization rate and nitrogen application rate are key factors affecting crop carbon emissions [29]. The R&D intensity and per capita disposable income of rural residents have a mitigating effect on agricultural carbon emissions, while the proportion of the agricultural labour force, agricultural added value, agricultural industrial structure and per capita arable land area promote increases in agricultural carbon emissions [30]. Life cycle assessment is the main research method used to study carbon effects of urban agriculture. On the one hand, the carbon footprint of agricultural and food sectors was evaluated from the perspective of the dietary consumption structure of urban residents [31]. On the other hand, a conventional small householder farm and large home-delivery agriculture were evaluated and compared from the two aspects of carbon footprint and economic efficiency, providing a reference for the selection of urban agricultural production and operation modes more conducive to carbon emission reduction [32,33].

At present, there are a series of studies mainly focusing on the quantitative evaluation of agricultural carbon effects. Little consideration is given to the stages of agricultural development and the evolution process of carbon effects after urban agriculture turns to multifunctionality. The influencing factors of carbon emissions have been considered; however, whether the same factors affect carbon sequestration and whether carbon emissions affect them in turn remain to be determined. As a megacity with a population of nearly 20 million, Guangzhou is faced with multiple pressures of protecting agricultural and ecological land while undergoing rapid urbanization. It is one of the classic cases of multifunctional urban agriculture research. Therefore, based on the changes in carbon emissions and carbon sequestration of urban agriculture in Guangzhou from 2002 to 2020, the Granger causality analysis method was used to investigate the interaction between multifunctionality and carbon effects of urban agriculture, and then the grey association model was used to analyse the agricultural development stages. Finally, carbon effects of urban agriculture in Guangzhou were analysed to provide background for the implementation of the "double carbon" strategy in metropolitan areas. From the perspective of research, this paper takes the agricultural transformation of the typical metropolis, Guangzhou, and uses it as the basis for a study of carbon effects. We explain changes in the carbon footprint in the context of the practicalities of urban agricultural development in Guangzhou, which broadens the research perspective in terms of space, time and effect. In terms of research content, this paper emphatically notes that the multifunctional transformation of urban agriculture can be divided into different stages, where the production, economic, social and ecological functions played by urban agriculture were different, and different carbon effects

were generated. This enriches the exploratory research on the ecological effects of regional agricultural systems and contributes to the search for carbon emission reduction strategies in metropolitan areas. In terms of research methods, this paper applies the Granger causality analysis method and the grey association model to explore the long-term time series interaction relationship between urban agricultural multifunctionality and carbon effects, which is suitable and innovative.

## 2. Materials and Methods

### 2.1. Study Area

Guangzhou is the capital of Guangdong Province in China, and it is located at 112°57′ E–114°3′ E, 22°26′ N–23°56′ N, with an administrative area of 7434.40 km$^2$, including 11 administrative regions: Liwan District, Yuexiu District, Haizhu District, Tianhe District, Baiyun District, Huangpu District, Panyu District, Huadu District, Nansha District, Conghua District and Zengcheng District (Figure 1). There are no agricultural production activities in Yuexiu District, which is not covered in this paper. From 2002 to 2020, the area of cultivated land in Guangzhou decreased from $1.1 \times 10^5$ hectares to $8.8 \times 10^4$ hectares, and the total agricultural output value increased rapidly from CNY 10.3 billion to CNY 51.4 billion. The per capita annual disposable income of rural residents increased from CNY 6857 to CNY 31,266. In 2020, the total income of urban agriculture in Guangzhou was CNY 261.3 billion. There were 1.55 million urban agricultural employees and 374 leading agricultural enterprises. Guangzhou needs to take the sustainability of agricultural production modes into account while ensuring the total agricultural output value to avoid the huge ecological pressure caused by operation modes with high carbon emissions. Urban agriculture is an important development direction of agriculture in the future. It is of great practical significance to study the evolution of carbon effects in the multifunctional transformation of urban agriculture in Guangzhou.

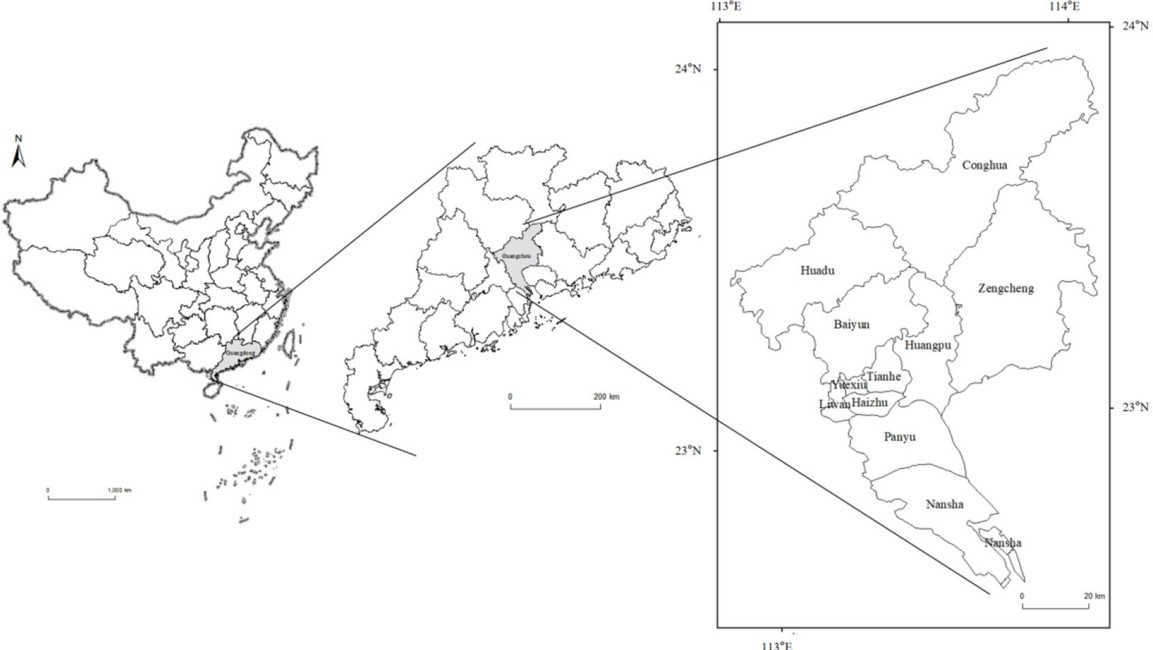

**Figure 1.** Location of Guangzhou, China.

### 2.2. Data Sources

The data for Guangzhou from 2002 to 2020 are from Guangzhou Statistical Yearbook (2003–2021). Net primary productivity (NPP) is the organic dry matter yield of green plants per unit time and area after subtracting autotrophic respiration. NPP data for 2002–2020 are from MOD17A3HGF Version (https://lpdaac.usgs.gov/product_search/?view=listhttps:

//lpdaac.usgs.gov/product_search/?vie6.0w=list, accessed on 18 Match 2022). Data on land use from 2002 to 2020 are from the 30 m annual land cover dataset in China (https://zenodo.org/record/4417810#.YShGWugzbBU, accessed on 18 Match 2022). The total annual PM2.5 data are from the Atmospheric Composition Analysis Group at Washington University (https://sites.wustl.edu/acag/datasets/surface-pm2--5/, accessed on 10 August 2021). The NDVI data are from the website of the Resources and Environmental Science and Data Center (https://www.resdc.cn/, accessed on 11 August 2021).

*2.3. Methods*

This paper aims to address the following scientific questions: (1) What kind of multifunctional transformation process did urban agriculture experience in Guangzhou? (2) What kind of carbon effects did urban agriculture generate in Guangzhou? (3) Is there an interaction between urban agricultural multifunctionality and carbon effects? How strong is the impact intensity? (4) What were the temporal characteristics of the interaction between urban agricultural multifunctionality and carbon effects? This paper follows the research logic of "process–effect–mechanism" and makes the following research design (Figure 2). First, the agricultural transformation process was displayed by constructing an evaluation index system of urban agricultural multifunctionality. At the same time, the continuous change process of emissions and sequestration generated by carbon sources and carbon sinks was examined by calculation. Then, the Granger causality analysis method was applied to test the interaction between the two, and an impulse response function was used to examine the impact intensity. Next, the grey association model was used to reveal the temporal characteristics of the interaction between the two. Finally, carbon effects generated by the multifunctional transformation of urban agriculture were explained according to the practicalities of urban agricultural development in Guangzhou on the basis of the literature.

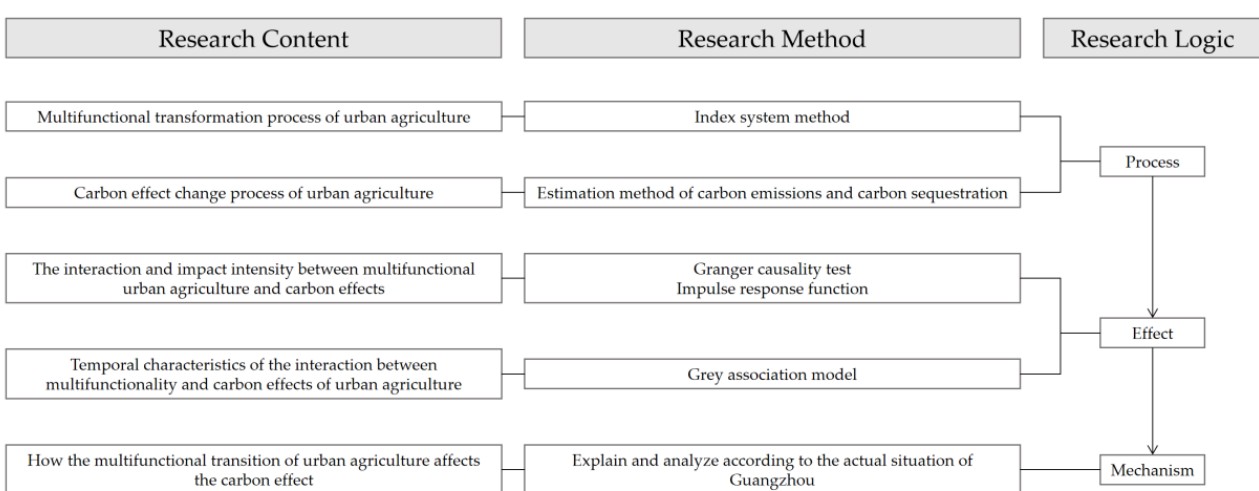

**Figure 2.** Study design.

2.3.1. Construction of Multifunctional Index System for Urban Agriculture

The development goal of the regional land "production–living–ecological" space is "intensive and efficient production space, habitable and appropriate living space, and ecological space with pure and natural beauty". This conceptual framework covers biophysical processes, direct and indirect production, as well as spiritual, cultural, leisure and aesthetic needs [34]. Based on the ideological connotation of the "three living" space, the index system of urban agricultural functions, namely production, economy, society and ecology, was constructed (Table 1). The production and supply function of urban agriculture is to ensure the quantity safety of food by increasing the diversification of food supply channels and increasing the self-reliance capacity of local food [35]. To ensure the quality safety of food [36], urban agriculture produces high-quality pollution-free agricultural

products [37] and provides fresh agricultural products and seasonal vegetables, which is the most basic function of urban agriculture. In terms of economic development functions, urban agriculture makes full use of urban talent and technology, promotes the upgrading of industrial structure [38], improves the output and value of agricultural products [39], and promotes the improvement of labour productivity and land productivity. In terms of social security functions, urban agriculture retains the production and farming characteristics of traditional agriculture; develops agricultural product processing, circulation and related industries by using a variety of agricultural resources; widens the channels for farmers to become rich; provides agriculture-related employment opportunities for migrant workers and the surplus labour force from other industries; and promotes the integration of groups with low labour skills into cities. In terms of ecological functions, urban agriculture can regulate urban climate, conserve water and soil, fix nitrogen and release oxygen [40], protect biodiversity, build urban green ecological barriers, curb urban sprawl [35], reserve urban development space [41], and promote harmony between human beings and nature [42]. The weight of each index was obtained by the entropy weight method.

**Table 1.** Assessing the index system of urban agricultural functions.

| Function | Index | Calculation Formula | Impact | Weight |
|---|---|---|---|---|
| Production Function | Cultivation index | Area of cultivated land/Land area | + | 8.61% |
| | Grain crop output per unit area | Yield of grain crops/Sown area of grain crops | + | 5.33% |
| | Per capita share of grain crops | Yield of grain crops/Permanent population at year-end | + | 40.94% |
| | Per capita share of fruits and vegetables | (Gross output of fruits + Yield of vegetables)/Permanent population at year-end | + | 26.18% |
| | Per capita share of agricultural products in animal husbandry | (Output of milk + Output of poultry eggs + Output of meat)/Permanent population at year-end | + | 18.94% |
| Economic Function | Agricultural output value per capita | Gross output value of agriculture, forestry, animal husbandry and fishery/Permanent population at year-end | + | 21.11% |
| | Proportion of gross output value of agriculture, forestry, animal husbandry and fishery | Gross output value of agriculture, forestry, animal husbandry and fishery/Gross domestic product | + | 25.54% |
| | Cultivated land productivity | Gross output value of agriculture, forestry, animal husbandry and fishery/Area of cultivated land | + | 25.92% |
| | Agricultural labour productivity | Gross output value of agriculture, forestry, animal husbandry and fishery/Total number of employed persons at year-end | + | 26.64% |
| | Rate of commodity output value of agriculture, forestry, animal husbandry and fishery | Commodity output value of agriculture, forestry, animal husbandry and fishery/Gross output value of agriculture, forestry, animal husbandry and fishery | + | 0.79% |
| Social Function | Per capita income level of rural residents | Per capita annual disposable income of rural residents | + | 29.45% |
| | Employment structure level | Number of rural employed persons in agriculture, forestry, animal husbandry and fishery/Total number of employed persons at year-end | + | 38.05% |
| | Agricultural service level | Proportion of service industry for agriculture in gross output value of agriculture, forestry, animal husbandry and fishery | + | 32.5% |
| Ecological Function | Vegetation coverage | Average of NDVI | + | 49.29% |
| | Air quality level | Total annual PM2.5 | - | 23.74% |
| | Degree of farmland fragmentation | Average of PD | - | 26.97% |

To eliminate the influence of dimension, nature difference and order of magnitude among indicators, the range standardization method was adopted. The standardization method of positive indicators is shown in Formula (1), and the standardization method of negative indicators is shown in Formula (2).

$$x_{ij} = (X_{ij} - X_{jmin})/(X_{jmax} - X_{jmin}) \tag{1}$$

$$x_{ij} = (X_{jmax} - X_{ij})/(X_{jmax} - X_{jmin}) \tag{2}$$

where $X_{ij}$, $X_{jmin}$, $X_{jmax}$, $x_{ij}$ are the original value, minimum value, maximum value and standardized value of the *j*-th index in the *i*-th area, respectively.

The calculation formula of each function score of each administrative region is as follows:

$$s_i = \sum_{j=1}^{m} x_{ij} w_j \tag{3}$$

where $s_i$ is the score of each function; $w_j$ is the weight of the *j*-th index; and *m* is the number of functional indicators.

### 2.3.2. Estimation Method of Carbon Emissions and Carbon Sequestration

The main sources of carbon emissions from agricultural inputs are pesticides, agricultural film, chemical fertilizers, agricultural machinery, agricultural irrigation and farmland tillage. The formula for estimating agricultural carbon emissions is as follows:

$$E = \sum E_i = \sum (T_i \times Q_i) \tag{4}$$

where E represents the total carbon emissions from agriculture; $E_i$ represents the carbon emissions of the *i*-th carbon source; $T_i$ represents the amount of the *i*-th carbon source; and $Q_i$ represents the carbon emission coefficient of the *i*-th carbon source (Table 2).

**Table 2.** Carbon emission coefficient.

| Carbon Source | Carbon Emission Coefficient |
| --- | --- |
| Agricultural pesticides | 4.9341 kg(C)·kg$^{-1}$ |
| Plastic film in agriculture | 5.18 kg (C)·kg$^{-1}$ |
| Chemical fertilizers | 0.8956 kg(C)·kg$^{-1}$ |
| Agricultural irrigation | 266.48 kg(C)·hm$^{-2}$ |
| Farmland tillage | 312.6 kg(C)·hm$^{-2}$ |
| Diesel oil in agriculture | 0.5927 kg(C)·kg$^{-1}$ |
| Agricultural ploughing | 16.47 kg(C)·hm$^{-2}$ |
| Agricultural electricity conversion | 0.18 kg(C)·kw$^{-1}$ |

Note: These data are from the carbon emission coefficient released by the IPCC.

The carbon sequestration of cultivated land can be obtained by adding the NPP corresponding to cultivated land in land use data with the ArcGIS grid calculator. According to the natural breakpoint method, the carbon emissions and carbon sequestration can be divided into five levels, namely low value area, medium-low value area, median area, medium-high value area and high value area in ascending order.

### 2.3.3. Granger Causality Test

The Granger causality test was used to investigate the relationship between carbon effects and urban agricultural multifunctionality. It is originally defined that if the lag value of one variable helps predict another variable, then that variable is the cause of the other variable. There are two time series $\{x_t\}$ and $\{y_t\}$. If:

$$x_t = \sum_{i=1}^{\infty} \alpha_i x_{t-i} + \sum_{i=1}^{\infty} \beta_i y_{t-i} + \varepsilon_i \tag{5}$$

$y_{t-i}$, the past value of $y$ is helpful to predict $x$, that is, there is at least one $i_0$, which makes $\beta_{i0} \neq 0$. Then the variable $y$ is the Granger cause of $x$. Before the panel Granger causality test, a unit root test needs to be performed on the variables to determine its stability, and then the cointegration relation of stationary series needs to be determined. To more intuitively analyse the impact of one endogenous variable on other endogenous variables, the impulse response function was used to describe the dynamic interaction between variables in the short term after Granger causality analysis.

### 2.3.4. Grey Association Model

The grey association model was used to quantitatively study the temporal characteristics of the interaction between multifunctionality and carbon effects of urban agriculture by drawing on the existing literature [43–45]. For the two dimensionless sequences $\{x_i\}$ and $\{y_j\}$, the associative coefficient and associative degree are calculated by Formulas (6) and (7):

$$\zeta_{ij}(t) = \frac{\overset{minmax}{i\quad j}\left|x_i(t) - y_j(t)\right| + \rho\overset{minmax}{i\quad j}\left|x_i(t) - y_j(t)\right|}{\left|x_i(t) - y_j(t)\right| + \rho\overset{minmax}{i\quad j}\left|x_i(t) - y_j(t)\right|} \tag{6}$$

$$\gamma_{ij} = \frac{1}{k}\sum_{i=1}^{k}\zeta_{ij}(t) \quad (k = 1, 2, 3, \ldots, n) \tag{7}$$

where $\zeta_{ij}(t)$ is the associative coefficient of the two indices at time t (space unit); $\rho$ is the discrimination coefficient, with the value in (0, 1). The smaller $\rho$ is, the greater difference between the correlation coefficients is, and the stronger the discrimination ability is. Generally take $\rho = 0.5$; k is the length of the time series; $\gamma_{ij}$ is the associative degree of the two indicators ($0 < \gamma_{ij} < 1$). The larger the value is, the greater the association is, and the more obvious the coupling effect between indicators is.

## 3. Results

### 3.1. Multifunctional Transformation Process of Urban Agriculture in Guangzhou

From 2002 to 2006, at the initial stage of urban agricultural development, the average value of the production function was 0.521, while the average values of the economic function and social function were 0.357 and 0.358, respectively. The production function was far stronger than other functions. From 2007 to 2015, with the accelerated development of urbanization, the urban demand for agricultural products and services changed. The economic function of agriculture in Guangzhou increased steadily from 0.369 to 0.400, and the social function saw a fluctuating increase from 0.377 to 0.409, with the average annual growth rate of both being 0.4%. At the same time, the production function decreased from 0.459 to 0.403, with an annual change rate of 0.7%. The development level of economic and social function gradually tended to be equal to the production function. From 2016 to 2020, with the development of urban agriculture becoming more mature, the economic function continued to increase from 0.387 to 0.442, with an annual average change rate of 1.4%. The social function continued to increase from 0.440 to 0.482, with an annual average change rate of 1.1%. The production function continued to decrease from 0.396 to 0.327, with an annual average change rate of 1.7%. The economic and social functions exceeded the production function, and urban agricultural multifunctionality was fully manifested (Figure 3). The ecological function was always weak, rising first and then decreasing from 2002 to 2020, with an average value of 0.106.

### 3.2. Carbon Effects in the Process of Multifunctional Transformation of Urban Agriculture

Urban agriculture in Guangzhou is a huge carbon sink as a whole (Figure 4). From 2002 to 2020, the average carbon sequestration was $1.1 \times 10^9$ kg, and the average carbon emissions were $2.1 \times 10^8$ kg. The carbon sequestration of urban agriculture in Guangzhou was approximately four times more than the carbon emissions. From 2002 to 2013, carbon

sequestration decreased from $1.2 \times 10^9$ kg to $1.0 \times 10^9$ kg, with an average annual change rate of 1.5%, and then increased steadily to $1.1 \times 10^9$ kg in 2020, with an average annual change rate of 1.4%. From 2002 to 2007, carbon emissions decreased from $2.4 \times 10^8$ kg to $1.9 \times 10^8$ kg, with an average annual change rate of 4.2%, and then increased to $2.2 \times 10^8$ kg from 2007 to 2010, with an average annual change rate of 5.3%. From then on to 2016, the value of carbon emissions remained at $2.2 \times 10^8$ kg and finally decreased to $1.9 \times 10^8$ kg from 2016 to 2020, with an average annual change rate of 3.4%.

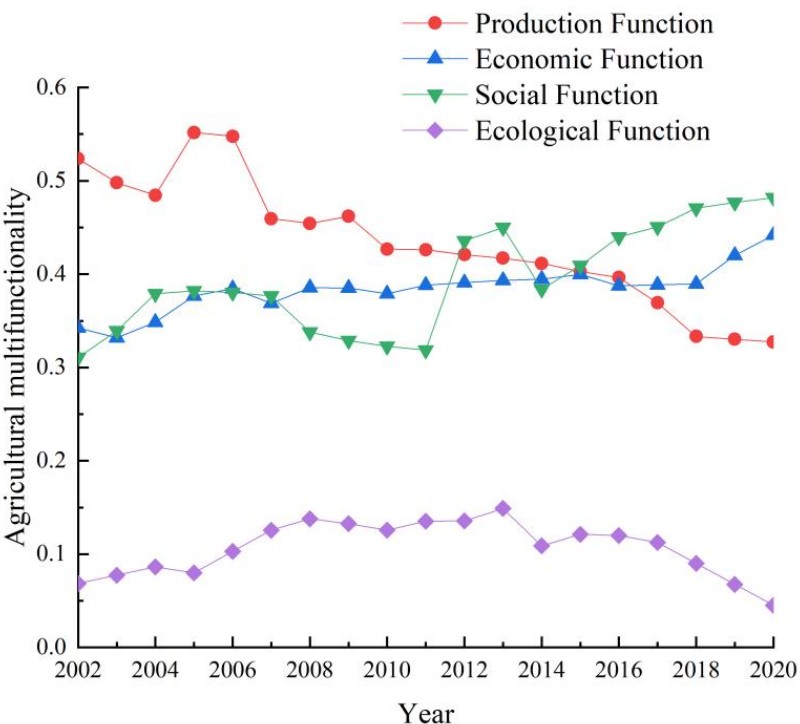

**Figure 3.** Urban agricultural multifunctionality in Guangzhou from 2002 to 2020.

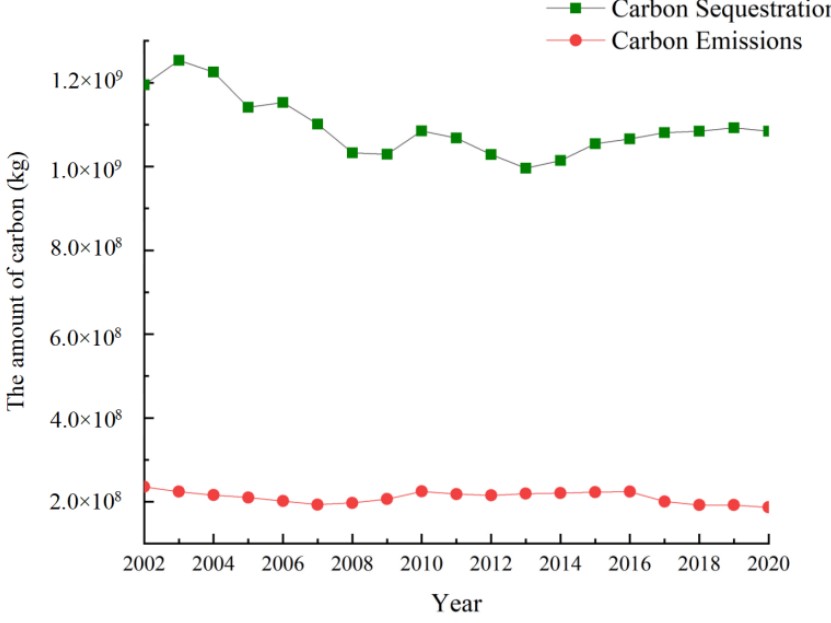

**Figure 4.** The carbon emissions and sequestration of urban agriculture in Guangzhou from 2002 to 2020.

From the perspective of temporal evolution, the carbon emissions of Panyu District, Baiyun District and Huadu District showed an overall downward trend, with decreases of $5.8 \times 10^7$ kg, $2.1 \times 10^7$ kg and $1.3 \times 10^7$ kg, respectively, from 2002 to 2020. The carbon emissions of Nansha District first increased significantly from $1.8 \times 10^7$ kg in 2005 to $5.8 \times 10^7$ kg in 2016 and then decreased slightly to $3.6 \times 10^7$ kg in 2020. The carbon emissions of Conghua District and Zengcheng District both showed a slight decrease at first and then a marginal increase, fluctuating at approximately $4 \times 10^7$ kg (Figure 5a). The carbon sequestration of each district was generally stable from 2002 to 2020. The average carbon sequestration from 2002 to 2020 was $4.8 \times 10^7$ kg in Panyu District, approximately $1.0 \times 10^8$ kg in Baiyun District, Huangpu District and Nansha District, $1.7 \times 10^8$ kg in Huadu District, and approximately $3 \times 10^8$ kg in Conghua District and Zengcheng District (Figure 5b). Liwan District, Haizhu District and Tianhe District were engaged in few agricultural activities, so carbon emissions and carbon sequestration were almost zero.

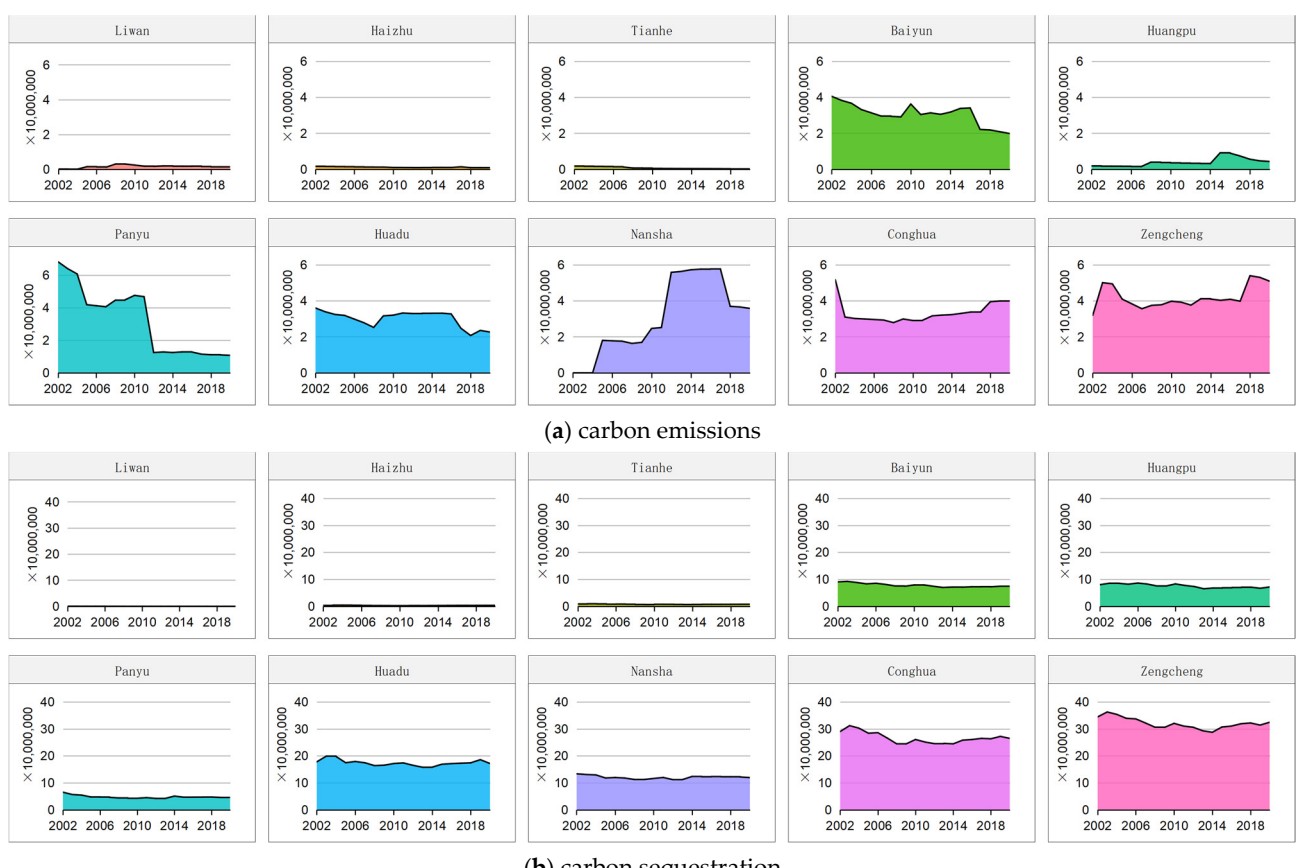

**Figure 5.** Carbon emissions and sequestration of urban agriculture in Guangzhou's districts from 2002 to 2020.

From the perspective of spatial distribution, the amount of agricultural production activities led to carbon emissions (Figure 6) and carbon sequestration (Figure 7) showing the characteristics of "core-edge". From 2002 to 2014, with the exception of no agricultural production in Yuexiu District, the core areas were Liwan, Haizhu, Tianhe and Huangpu, where agricultural carbon emissions were at the low value area, ranging from $5.7 \times 10^4$–$3.9 \times 10^6$ kg. Baiyun, Huadu, Panyu, Nansha, Conghua and Zengcheng were marginal areas, forming a clear boundary with the core areas, whose agricultural carbon emissions were at or above the median area, with a range of $1.2 \times 10^7$–$6.8 \times 10^7$ kg. The gap from core to edge was large. From 2015 to 2016, Liwan, Haizhu and Tianhe were still the core areas. Huangpu and Panyu became the sub marginal areas, and the agricultural carbon emissions were at the medium-low value area, ranging from $8.9 \times 10^6$–$1.3 \times 10^7$ kg.

Baiyun, Huadu, Nansha, Conghua and Zengcheng were marginal areas, where agricultural carbon emissions were at the medium-high value area or high value area, ranging from $3.3 \times 10^7$–$5.8 \times 10^7$ kg. In 2017, Baiyun and Huadu changed from marginal areas to sub marginal areas, where agricultural carbon emissions changed from medium-high value areas to median areas. From 2018 to 2020, the core areas were Liwan, Haizhu, Tianhe and Huangpu. Baiyun, Huadu and Panyu were sub marginal areas, where agricultural carbon emissions were at the medium-low value area, ranging from $1.1 \times 10^7$ to $2.4 \times 10^7$ kg. Nansha, Conghua and Zengcheng were marginal areas, where agricultural carbon emissions were at the medium-high value area or high value area, ranging from $3.6 \times 10^7$ to $5.4 \times 10^7$ kg. The gap from core to edge gradually shrank. The "core-edge" pattern of carbon sequestration was similar to that of carbon emissions, with little change in each district.

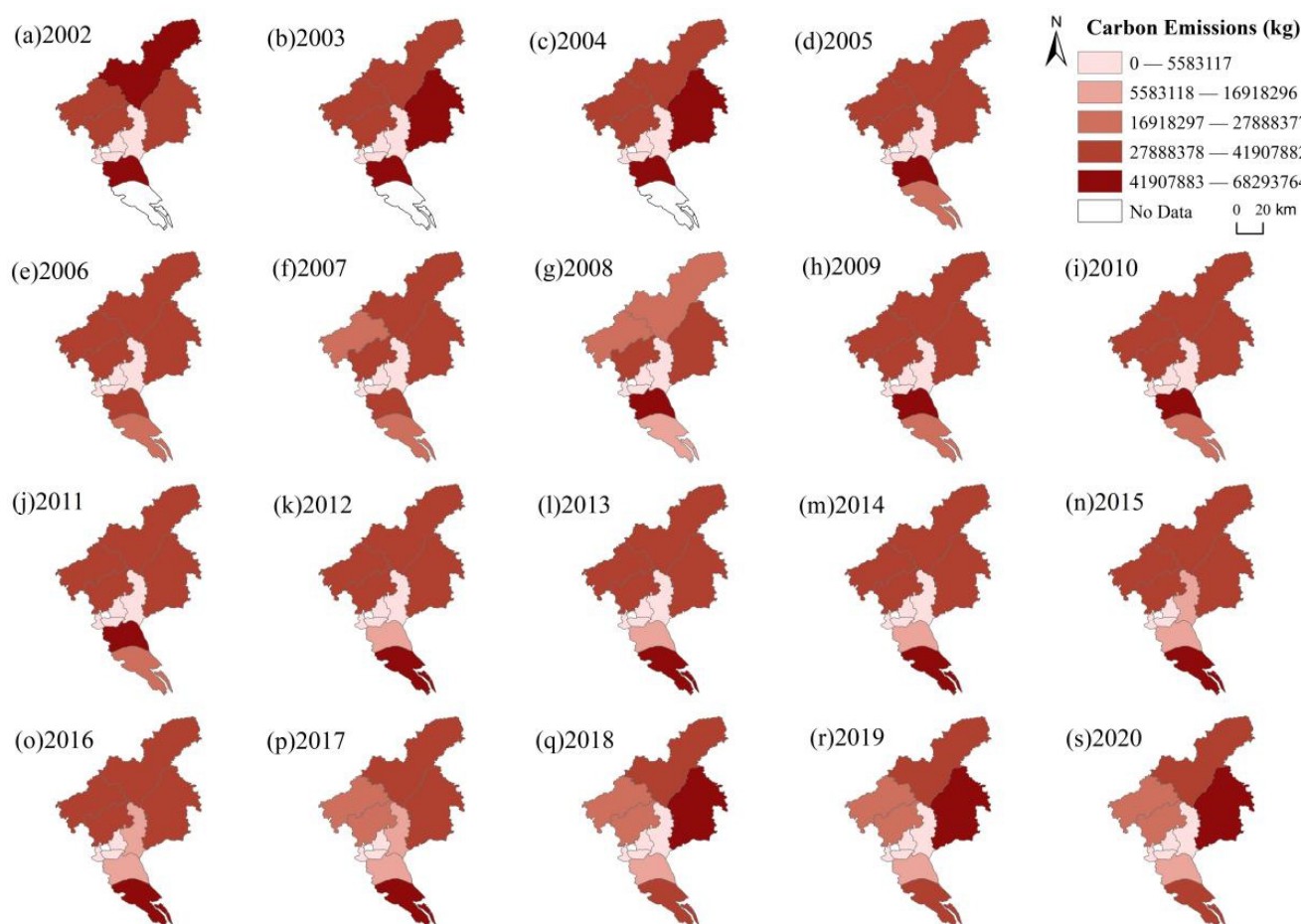

**Figure 6.** Carbon emissions of urban agriculture in Guangzhou's districts from 2002 to 2020.

### 3.3. The Causal Test of Multifunctional Transformation and Carbon Effects of Urban Agriculture

The ADF unit root test method was used to test the stability of the carbon effect and agricultural multifunctional variables (Table 3). The results show that when only the constant term is included, the $p$ values of carbon emissions, carbon sequestration, production function, economic function, social function and ecological function are all less than 0.05, which leads to rejecting the null hypothesis of "existence of unit root", and the variables are determined to be stationary series. Therefore, all variables are single-integration sequences of the same order and have conditions for further cointegration testing.

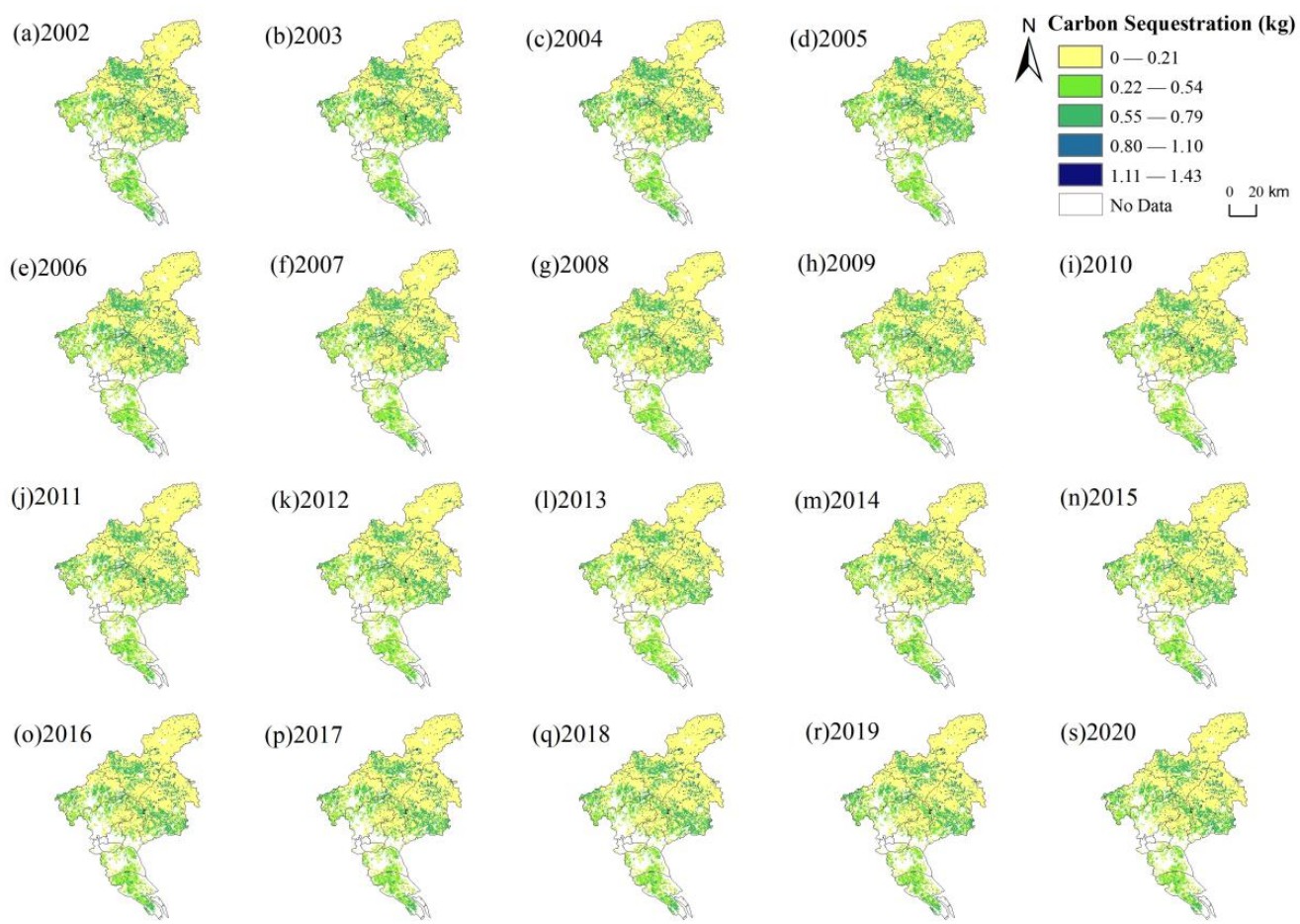

**Figure 7.** Carbon sequestration of urban agriculture in Guangzhou's districts from 2002 to 2020.

**Table 3.** Results of panel unit root tests.

| Variable | Test Type (C,T,K) | *p* Value | Result |
|---|---|---|---|
| Carbon emissions | C,0,0 | 0.0000 | Stationary |
| Carbon sequestration | C,0,0 | 0.0000 | Stationary |
| Production function | C,0,0 | 0.0000 | Stationary |
| Economic function | C,0,0 | 0.0000 | Stationary |
| Social function | C,0,0 | 0.0000 | Stationary |
| Ecological function | C,0,0 | 0.0000 | Stationary |

Note: C, T and K in test types represent the constant term, trend term and lag order, respectively. A value of 0 means that the test model does not contain a constant term or time trend term, or the lag order is 0.

The EG method based on residuals was adopted for the cointegration test. First, cointegration regression was performed on the following combinations of variables, and then the ADF test was performed on the residual sequence obtained (Table 4). The results show that when neither the constant term nor the trend term is included, the *p* values of the residual sequences of all variables are less than 0.05. Therefore, it can be determined that there is a cointegration relationship between carbon effects and agricultural multifunctionality.

The lag order was determined by the method of multicriteria joint determination, where the LR, FPE, AIC, SC and HQ criteria were included. On this basis, the Granger causality test was conducted to analyse the causal relationship between the combinations of variables (Table 5). The results show that the carbon emissions of urban agriculture in Guangzhou have a causal relationship with the production, social and ecological functions. Carbon emissions are the Granger cause of the economic function but not the opposite. The carbon sequestration of urban agriculture in Guangzhou has a causal relationship with the production and economic function. Carbon sequestration is the Granger cause of the

ecological function but not the opposite. There is no Granger causality between carbon sequestration and the social function when the lag order is eight.

**Table 4.** Results of the panel cointegration test.

| Variable | Test Type (C,T,K) | *p* Value | Result |
|---|---|---|---|
| Carbon emissions and Production function | 0,0,0 | 0.0000 | There exists cointegration. |
| Carbon emissions and Economic function | 0,0,0 | 0.0000 | There exists cointegration. |
| Carbon emissions and Social function | 0,0,0 | 0.0000 | There exists cointegration. |
| Carbon emissions and Ecological function | 0,0,0 | 0.0000 | There exists cointegration. |
| Carbon sequestration and Production function | 0,0,0 | 0.0000 | There exists cointegration. |
| Carbon sequestration and Economic function | 0,0,0 | 0.0000 | There exists cointegration. |
| Carbon sequestration and Social function | 0,0,0 | 0.0000 | There exists cointegration. |
| Carbon sequestration and Ecological function | 0,0,0 | 0.0000 | There exists cointegration. |

Note: C, T and K in test types represent the constant term, trend term and lag order, respectively. A value of 0 means that the test model does not contain a constant term or time trend term, or the lag order is 0.

**Table 5.** Results of Granger causality test.

| Variable | Lag Order | *p* Value | Result |
|---|---|---|---|
| Carbon emissions → Production function | 8 | 0.0326 | There exists Granger causality. |
| Production function → Carbon emissions | 8 | 0.0058 | There exists Granger causality. |
| Carbon emissions → Economic function | 8 | 0.0011 | There exists Granger causality. |
| Economic function → Carbon emissions | 8 | 0.5414 | There exists no Granger causality. |
| Carbon emissions → Social function | 7 | 0.0000 | There exists Granger causality. |
| Social function → Carbon emissions | 7 | 0.0000 | There exists Granger causality. |
| Carbon emissions → Ecological Function | 8 | 0.0081 | There exists Granger causality. |
| Ecological Function → Carbon emissions | 8 | 0.0479 | There exists Granger causality. |
| Carbon sequestration → Production function | 6 | 0.0000 | There exists Granger causality. |
| Production function → Carbon sequestration | 6 | 0.0000 | There exists Granger causality. |
| Carbon sequestration → Economic function | 6 | 0.0005 | There exists Granger causality. |
| Economic function → Carbon sequestration | 6 | 0.0038 | There exists Granger causality. |
| Carbon sequestration → Social function | 8 | 0.0936 | There exists no Granger causality. |
| Social function → Carbon sequestration | 8 | 0.5513 | There exists no Granger causality. |
| Carbon sequestration → Ecological Function | 8 | 0.0005 | There exists Granger causality. |
| Ecological Function → Carbon sequestration | 8 | 0.7917 | There exists no Granger causality. |

All functions of urban agriculture do not immediately respond to the disturbance from carbon emissions and carbon sequestration, and the response value of the first phase is 0 (Figure 8a,c,e,g,i,k,m,o). The effect of carbon emissions on the production function (Figure 8a) is very similar to that of the production function on carbon emissions (Figure 8b). The mean values of impact intensity are both approximately 0.016, and the directions of impact are roughly opposite in each lag period. The effect of carbon emissions on economic function (Figure 8c) is similar to that of the economic function on carbon emissions (Figure 8d). The mean impact intensities are 0.040 and 0.034, respectively, and the directions of impact are roughly opposite. The effect of carbon emissions on the social function (Figure 8e) and the effect of the social function on carbon emissions (Figure 8f) are generally similar, with the mean values of impact intensity being 0.008 and 0.011, respectively. The difference is that the social function is more susceptible to the effect of carbon emissions in the early stage and less susceptible in the later stage, while the effect of the social function on carbon emissions is small in the early stage and increases in the later stage. The effect of carbon emissions on the ecological function (Figure 8g) and that of the ecological function on carbon emissions (Figure 8h) are quite different, with the mean impact intensities of 0.013 and 0.004, respectively, while the directions are roughly the same. The effect of carbon sequestration on the production function (Figure 8i) is far from that of the production function on carbon sequestration (Figure 8j). The mean values of impact intensity are 0.015 and 0.269, respectively. The influence directions of the two are roughly

the same, with both swinging back and forth between positive and negative, and the impact strength increases gradually. The effect of carbon sequestration on the economic function (Figure 8k) and that of the economic function on carbon sequestration (Figure 8l) have great differences. The mean values of impact intensity are 1.035 and 2.778, with completely opposite directions. Both of them swing back and forth between positive and negative, and the impact intensity of the later stage is approximately twice that of the previous stage. The effect of carbon sequestration on the social function (Figure 8m) is far from that of the social function on carbon sequestration (Figure 8n). The mean values of impact intensity are 0.001 and 0.037, respectively, and the influence directions are roughly opposite. The effect of carbon sequestration on the ecological function (Figure 8o) and that of the ecological function on carbon sequestration (Figure 8p) are significantly different, with the mean impact intensities of 0.001 and 0.026, respectively, while the influence directions are roughly the same.

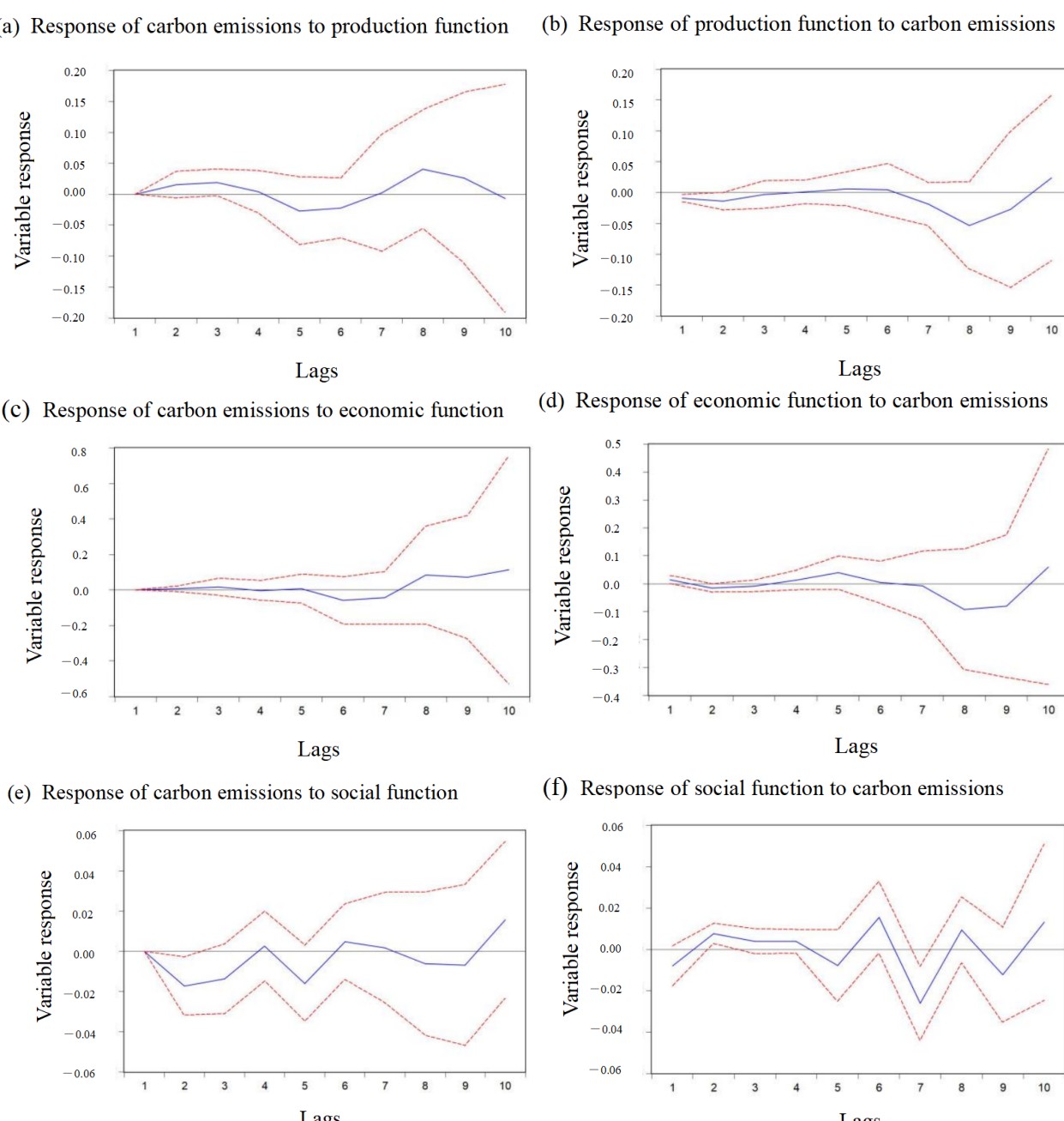

**Figure 8.** *Cont.*

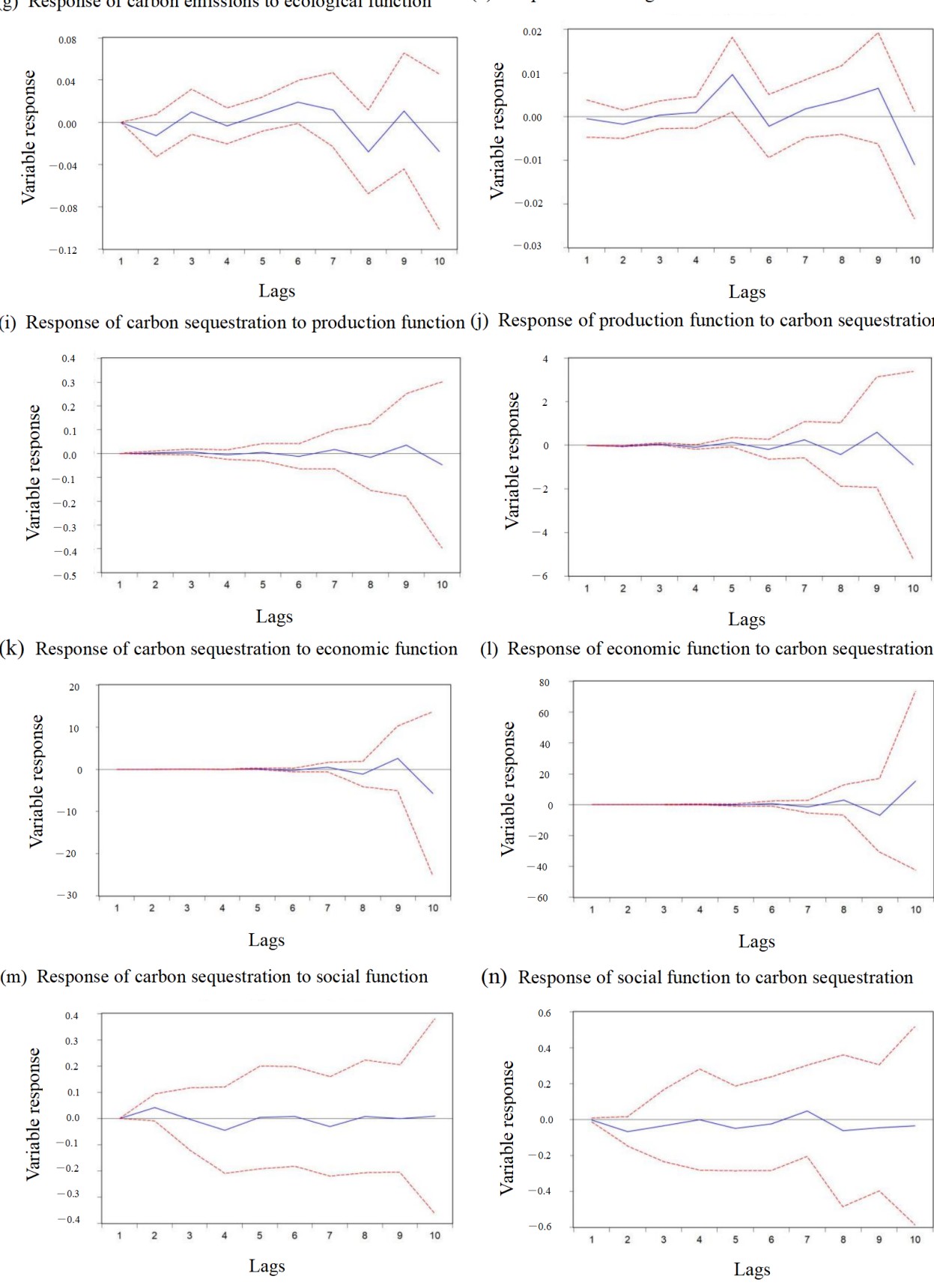

**Figure 8.** *Cont.*

(o) Response of carbon sequestration to ecological function    (p) Response of ecological function to carbon sequestration

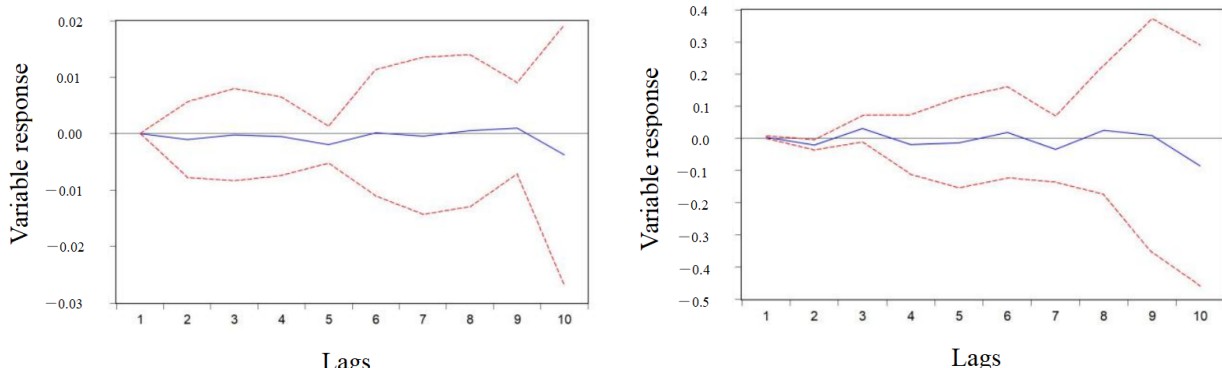

**Figure 8.** Impulse response function of carbon effects and agricultural multifunctionality.

### 3.4. Temporal Characteristics of the Associative Degree between Multifunctionality and Carbon Effects of Urban Agriculture

The associative degree between the production function and carbon emissions as well as carbon sequestration is the highest compared with other functions. The associative degree between the production function and carbon sequestration (0.862) is greater than that between the production function and carbon emissions (0.826), showing that the carbon sink is the role of the agricultural production system more than the carbon source. In contrast, the associative degree between economic function and carbon emissions (0.803) is greater than that between the economic function and carbon sequestration (0.786), showing that agriculture tends to act as a carbon source in the pursuit of economic benefits. The associative degree between the social function and carbon sequestration (0.796) is greater than that between the social function and carbon emissions (0.768). The role of agriculture as a carbon sink is stronger than that of a carbon source when it exerts social functions such as providing employment, wage and service products. The associative degree between the ecological function and carbon effects is the lowest, and its contribution to carbon effects is limited (Figure 9).

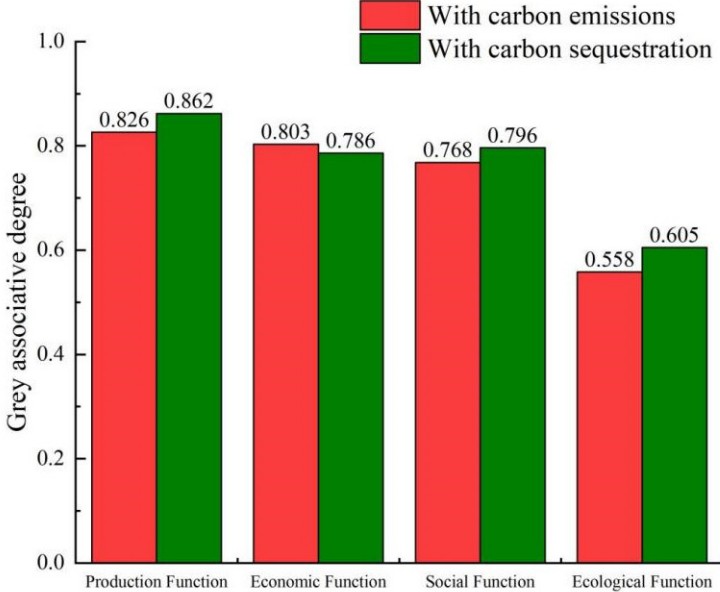

**Figure 9.** Grey associative degree between agricultural multifunctionality and carbon emissions as well as carbon sequestration.

According to the multifunctionality of urban agriculture in Guangzhou from 2002 to 2020 (Figure 3) and the change in its degree of association with carbon effects (Figure 10), the interactive development process of urban agricultural multifunctionality and carbon effects can be divided into three stages. The first stage was from 2002 to 2006 when agriculture in Guangzhou was dominated by the production function, and the economic and social functions were relatively weak. At this time, the associative degree between the economic function and carbon emissions was at a high level (the average associative degree between 2002 and 2006 was 0.813) and showed an increasing trend. The associative degree between the production function and carbon emissions (the average associative degree between 2002 and 2006 was 0.780) was less than that of the economic function. The associative degree between the social function and carbon emissions was lower (the average associative degree between 2002 and 2006 was 0.721). The associative degree between the ecological function and carbon emissions first decreased and then increased, with an average value of 0.549 (Figure 10a). The level of interaction between the agricultural production function and carbon sequestration was at a low stage (the average associative degree between 2002 and 2006 was 0.801). The associative degree between the economic and social functions and carbon sequestration was slightly lower than that of the production function, with the average values of 0.791 and 0.790, respectively. The associative degree between the ecological function and carbon sequestration first decreased and then increased, with an average value of 0.613 (Figure 10b).

The second stage was from 2007 to 2015. The production function declined steadily, and the economic and social functions gradually increased to the same level as the production function. At this time, the associative degree between the production function and carbon emissions showed an increasing trend, from 0.799 in 2007 to 0.861 in 2015. In contrast, the associative degree between the economic function and carbon emissions began to decline after reaching the peak at the end of the previous stage, from 0.837 in 2007 to 0.780 in 2015. The associative degree between the social function and carbon emissions fluctuated at a low level, with an average of 0.782 from 2007 to 2015. The associative degree between the ecological function and carbon emissions first increased and then decreased, with an average value of 0.583 (Figure 10a). The interaction level between the agricultural production function and carbon sequestration was at a high stage (the average associative degree between 2007 and 2015 was 0.884). The associative degree between the economic function and carbon sequestration showed a downward trend at this stage, from 0.821 in 2007 to 0.774 in 2015. The average associative degree between the social function and carbon sequestration was 0.804 in 2007–2015. The associative degree between the ecological function and carbon sequestration first increased and then decreased, with an average value of 0.627 (Figure 10b).

The third stage was from 2016 to 2020. The production function further declined, and the economic and social functions exceeded the production function. Urban agricultural multifunctionality fully manifested. At this time, the associative degree between the production function and carbon emissions (the average associative degree between 2016 and 2020 was 0.849) was always greater than that of the economic function (the average associative degree between 2016 and 2020 was 0.803) and social function (the average associative degree between 2016 and 2020 was 0.782). The associative degree between the ecological function and carbon emissions increased steadily, with an average value of 0.622 (Figure 10a). The associative degree between the agricultural production function and carbon sequestration decreased slightly and steadily at this stage, from 0.906 in 2016 to 0.855 in 2020. The associative degree between the economic function and carbon sequestration further decreased from 0.794 in 2016 to 0.740 in 2020. The associative degree between the social function and carbon sequestration fluctuated slightly, with an average value of 0.789. The associative degree between the ecological function and carbon sequestration decreased steadily, with an average value of 0.616 (Figure 10b).

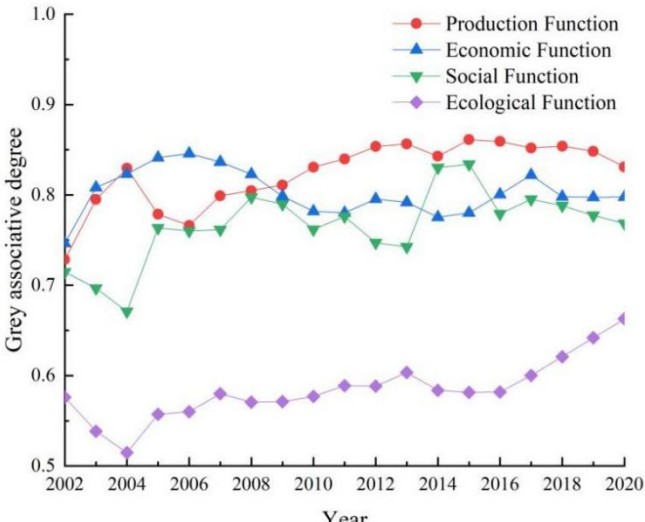

(**a**) Grey associative degree between carbon emissions and agricultural multifunctionality

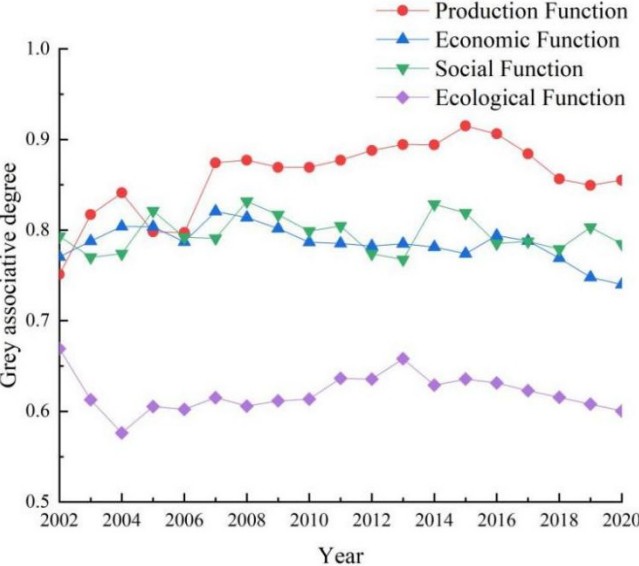

(**b**) Grey associative degree between carbon sequestration and agricultural multifunctionality

**Figure 10.** Grey associative degree between agricultural multifunctionality and carbon emissions as well as carbon sequestration from 2002 to 2020.

## 4. Discussion

### 4.1. Carbon Emission Reduction Effect of Urban Agricultural Multifunctional Transformation

The carbon emission reduction function of urban agriculture is mainly considered from two aspects: low-carbon input and low-carbon output. The two are realized through low-carbon production modes and lifestyles, corresponding to the production and social functions of urban agriculture, respectively. It can be seen from the impulse response function that in the significant lag period, the production function (Figure 8b) and social function (Figure 8f) have a significant negative impact on carbon emissions, which can promote emission reduction, while the emission reduction effect of the economic function did not pass the significance test (Table 5).

In terms of the production function, emission reduction is mainly achieved through the technical improvement of carbon sources [46]. First, the conservation tillage system is implemented for soil tillage. Straw returning to the field and the rotation system of double cropping rice + winter planting green manure are promoted to reduce the physical disturbance of the soil and improve the stability. The fallow system is adopted for farmlands

with long-term cultivation and soil weathering and corrosion to reabsorb and store soil organic carbon. For fallow farmland, the storage capacity of soil organic carbon and soil fertility can be increased by covering vegetation on the surface again and effectively preserving crop residues [47]. At the same time, no or less tillage can control the carbon emissions generated by fossil fuel combustion by reducing the use of agricultural machinery. In addition, the increase in soil fertility can effectively reduce the use of chemical fertilizers in farming, thereby reducing carbon emissions [48]. Second, water-saving and drought-resistant technology is adopted for agricultural irrigation. Irrigation is carried out according to the critical period of crop growth to improve the effective utilization rate of irrigation water. Increasing soil organic matter can reduce soil water evaporation and improve the drought-resistance ability of soil. At the same time, drought-resistant varieties are selected and combined with the application of chemical drought-resistant agents [49]. Third, for the application of chemical fertilizer, precise agricultural production methods and UAV precision fertilization are adopted. People accurately estimate the fertilizer demand of crops and locate fertilization, applying nitrogen fertilizer at the position that is most easily absorbed by the roots of crops to improve its utilization level [50]. Fourth, for the application of pesticides, the unified control of crop diseases and pests is adopted to avoid large-scale spraying of pesticides [51]. Finally, the rational use of agricultural film, the development of waste agricultural film reuse technology and the replacement of plastic agricultural film with straw fibre agricultural film of natural products and agricultural and sideline products are advocated [48].

In terms of the social function, to achieve emission reduction, the government mainly improves the policy of strengthening grain and benefiting agriculture and promotes the operation mechanism of farmers' participation in the new agricultural enterprises, trustee-ship of agricultural production and integration of agriculture, culture and tourism industry to increase farmers' income and employment. The government spares no effort to build a "Trinity" support policy system of price, subsidy and insurance, adheres to and improves agricultural subsidy policies such as land circulation, large grain growers, rice insurance, agricultural machinery purchase and operation, achieving a high and stable yield of farm-land and guaranteed income in droughts and floods [52]. Interest consortia of "state-owned enterprises + village collectives + social enterprises" are established, and a new mechanism of linking agriculture is created. The relationship between enterprises and farmers turns from "game" to "cooperation", which encourages farmers to convert resources, assets, funds and technologies into shares and obtain dividend payments according to shares of industrial and land value-added income. Farmers achieve a multichannel income increase of "rent + share capital + salary", changing from a passive "participant" to a "partner" in agricultural development, fully enjoying the dividends brought by the rapid development of agriculture [53]. At the same time, the new agricultural machinery management mode of "agricultural nannies" is implemented, and the mechanized facilities that are needed in production, such as field climbing machines, rice transplanters, and UAVs are gathered to form an agricultural machinery service society, thereby realizing the full mechanization of farming, planting, management, and harvesting. This improves production efficiency and reduces the purchase pressure of agricultural machinery of family farms, large plant-ing households and other production and management entities [54]. In addition, urban agriculture in Guangzhou integrates the development of leisure, sightseeing, culture and tourism, education, science popularization and other industries, builds agricultural parks, and introduces the agroforestry economic and ecotourism complex with the integrated development of agriculture, culture and tourism industry according to local conditions, cre-ating a large number of jobs and driving the sales of catering, home stays and agricultural products [55]. Agricultural producers have opportunities to learn the low-carbon concept by participating in the demonstration, experiment and consultation activities carried out by technology popularization institutions, information institutions and agricultural machinery enterprises. Sustainability-oriented business models and development concepts are in-creasingly accepted and valued by people [56]. Farmers who have received comprehensive

decision-making training usually show a willingness to turn to social ecosystem thinking and conduct critical personal and professional self-reflection. They tend to have an inclusive and open attitude, participate in the supporting practice community and be willing to learn and strengthen the understanding and observation of resources such as soil and pasture [57]. The increase in farmers' income, the transformation in lifestyle, the improvement of modern agricultural knowledge and skills, and the increasing awareness of the safety and superiority of low-carbon agricultural products will help reduce carbon emissions.

*4.2. Carbon Sequestration Increase Effect of Urban Agricultural Multifunctional Transformation*

From the impulse response function, it can be seen that in the significant lag period, the economic function has a significant positive impact on carbon sequestration (Figure 8l), which has the effect of increasing carbon sequestration. In contrast, the production function has a negative impact on carbon sequestration (Figure 8j), which has a carbon sequestration reduction effect. The relationship between the social function and carbon sequestration is not significant (Table 5).

In terms of the economic function, the carbon sequestration increase effect is achieved mainly through the way that leading enterprises lead the integration of the three industries and through Internet sales channels opening up. In addition to the government's financial investment support [4], the development of a green economy requires the active involvement of financial capital and market capital [58]. For urban agriculture in Guangzhou, based on the agricultural investment group, subsidiaries are set up in subdistricts, which act as an effective bridge between the government and the market and are responsible for the specific implementation of regional investment cooperation and operation. National leading enterprises with strong innovation ability are introduced, and powerful state-owned and private enterprises are guided to take shares with brand, technology, assets and other factors [59]. A number of modern agricultural projects such as smart agriculture, digital agriculture and deep processing [60] have been introduced to build the 5G smart agricultural pilot zone and the global brand agricultural products trading and pricing centre [61], which creates a full chain digital industrial platform with "brain for planting, wisdom for growth, and traceability for sales" [62], integrates planting, warehousing, processing, and brand sales, and realizes ecology and intelligence [63]. The investment in the digital economy and the concept of the ecological economy enable business entities to obtain greater benefits from the promotion of low-carbon agricultural production methods or technologies [64]. Strong willingness and technical support promote research on carbon neutralization green production modes of brand agricultural products by measuring the carbon emissions and carbon sequestration of crops under different planting modes, determining the carbon sink base and exploring the optimal production mode of carbon sinks.

*4.3. A Long Time Lag between Multifunctional Transformation and Carbon Effects of Urban Agriculture*

Changes in production methods, agricultural policies and business models lead to the transformation of urban agricultural functions, and carbon effects of urban agriculture change accordingly. As seen from the impulse response function, the response of carbon effects to the multifunctional transformation of urban agriculture is weak in the early stage, and with the increase in lag period, the response intensity increases gradually (Figure 8). A significant impact relationship between the multifunctional transformation of urban agriculture and the change in carbon effects requires an interval of 6–8 years. There is a long time lag between multifunctional transformation and carbon effects of urban agriculture. The reasons for this long time lag are mainly the following two aspects. From the perspective of the multifunctional transformation of urban agriculture, the production processes from research to application, the agricultural policy processes from proposal to implementation and the business model processes from attempts to operation are all exploratory, thereby taking time. From the perspective of carbon effects, the process of

carbon cycle in the atmosphere, phytosphere and pedosphere is a long period, so the interaction between the two has an obvious time effect.

The main reason for the lag of the emission reduction effect of the production function is that crop growth and soil fertility recovery have a certain periodicity. Straw returning to the field means that straw is piled into the soil directly or after maturity, which can be converted into organic matter after a period of decomposition [65]. The rotation system is the alternating planting of flood and drought crops according to the season and soil environment, with obvious periodicity [66]. The cultivation of drought-resistant varieties requires a development cycle from gene implantation to character appearance [67]. Positioning and quantification in the process of irrigation, fertilization and spraying depends on the growth of crops, which also has a lag [68]. The lag of the emission reduction effect of the social function is mainly caused by the following two aspects. On the one hand, agricultural benefit policies are delayed from proposal to implementation [69]; on the other hand, farmers' acceptance of the low-carbon concept is not achieved overnight, but requires a certain time process [56]. It is a long-term process for farmers to recognize, participate in and benefit from the new management models. In this process, the transformation of farmers' production concepts and lifestyles to low carbon is gradual, so the emergence of emission reduction effects lags behind the social function in urban agriculture. As for the lag of the carbon sequestration increase effect on the economic function, it is also a process that needs continuous exploration from brand building, input of various business ingredients, feedback of sales, to the improvement of ecological organic varieties [70]. Only through repeated measurement and debugging can the optimal production mode of carbon sinks be finally obtained.

### 4.4. Uncertainty

The research scope of urban agriculture in this paper only includes the planting industry, without considering other agricultural types, such as animal husbandry, fishery and forestry, and $CO_2$ emissions are considered the only negative impact of agriculture on the environment. In fact, the production process of agriculture will also produce greenhouse gas emissions such as $CH_4$ and $N_2O$. Therefore, this paper does not comprehensively evaluate and analyse the impact of agriculture on the atmosphere and the reaction of climate change on agricultural production. In addition, in terms of the social function of urban agriculture, household characteristics such as household working population, the proportion of agricultural income in total income and the education level of household heads will have an impact on the decision making of whether farmers adopt low-carbon technology and then affect agricultural carbon effects. The impact of agricultural household demographic characteristics on carbon effects remains to be further studied.

### 5. Conclusions and Policy Enlightenment

Based on the changes in agricultural carbon emissions and carbon sequestration in Guangzhou from 2002 to 2020, we used the Granger causality analysis method to investigate the interaction between urban agricultural multifunctionality and carbon effects and then used the grey association model to analyse the evolution process of associative degrees between the two. We then divided the agricultural development into stages. Finally, the carbon effects produced in the process of multifunctional transformation of urban agriculture in Guangzhou were analysed. The conclusions are as follows. (1) From 2002 to 2020 in Guangzhou, urban agricultural production function decreased, the economic and social function increased, and the ecological function climbed and then declined. The carbon sequestration of urban agriculture in Guangzhou was approximately four times more than the carbon emissions. Carbon emissions experienced a process of first decreasing, then increasing, then remaining constant, and finally decreasing, while carbon sequestration first decreased and then increased. (2) The carbon emissions of urban agriculture in Guangzhou have a causal relationship with the production, social and ecological functions. Carbon emissions are the Granger cause of the economic function but not the opposite. The carbon

sequestration of urban agriculture in Guangzhou has a causal relationship with production and economic functions. Carbon sequestration is the Granger cause of the ecological function but not the opposite. There is no Granger causal relationship between carbon sequestration and the social function. (3) From 2002 to 2020, the interactive development process of urban agricultural multifunctionality and carbon effects in Guangzhou can be divided into three stages: production function oriented (2002–2006), economic and social function enhanced and production function weakened (2007–2015) and the economic and social function exceeded the production function (2016–2020). (4) The multifunctional transformation of urban agriculture has reduced carbon emissions and increased sequestration. There is a long time lag between multifunctional transformation and carbon effects of urban agriculture.

Compared with traditional rural agriculture and urban industrial production, urban agriculture has made positive contributions to carbon emission reduction and carbon sequestration due to its own particularity. According to the emission reduction and sequestration increase effect generated in the process of the multifunctional transformation of urban agriculture, the policy implications on how to solve the problem of excessive carbon dioxide emissions through agriculture in metropolitan areas are put forward as follows. First, by increasing efforts in technological research and development and changing the production process of soil tillage, irrigation, fertilization, spraying and covering agricultural film, carbon emissions are reduced. Second, innovating the operation mechanism of agricultural enterprises, realizing the integrated development of the three industries, building the cooperation platform of all stakeholders, making farmers benefit from the development of agriculture and the integration of industry, and providing technical training and demonstration consulting for farmers regularly is conducive to transforming their production and life concepts to low carbon. Third, deepening the reform of agricultural modernization, introducing modern agricultural projects such as digital economy and smart agriculture, opening up Internet sales channels, and guiding operators to broaden the industrial chain and create brand agricultural products under the logic of pursuing maximum profit promotes research on the carbon neutral green production mode of agricultural products.

**Author Contributions:** Conceptualization, Z.S.; methodology, Z.S.; formal analysis, Z.S.; investigation, Z.S.; data curation, Z.S.; writing—original draft preparation, Z.S.; writing—review and editing, R.Y.; visualization, Z.S.; supervision, R.Y.; project administration, R.Y.; funding acquisition, R.Y. All authors have read and agreed to the published version of the manuscript.

**Funding:** This work was supported by financial support from the Key-Area Research and Development Program of Guangdong Province (2020B0202010002), National Natural Science Foundation of China (41871177, 42171193, 41801088), Guangzhou Science and Technology Project (202102080254).

**Data Availability Statement:** The data are not publicly available for privacy reasons.

**Conflicts of Interest:** The authors declare no conflict of interest.

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
