# Peer review of "The Interaction and Its Evolution of the Urban Agricultural Multifunctionality and Carbon Effects in Guangzhou, China"

_land, doi:10.3390/land11091413_

Round 1

Reviewer 1 Report

         This is an interesting study. The authors focused on the impact of urban agricultural versatility on carbon emissions based on long time series data in Guangdong. In general, the research perspective is novel, the research design is reasonable, and the method analysis is appropriate. The research conclusions can provide decision-making reference for the formulation of relevant policies. It is suggested to be published after minor repair. The specific opinions are as follows:

         (1) The marginal contribution of research needs to be further clarified. It is suggested that the last paragraph of the introduction should systematically explain the marginal contribution of this research, for example, from the perspective of research, research content, research methods and other aspects to do a systematic discussion.

         (2) Why we should focus on the multifunctionality of urban agricultural may need to say more. Compared with the well-known contribution of traditional rural agricultural production and urban industrial production to carbon emissions, the functionality of urban agricultural multifunctionality that the author focuses on is special. That is the highlight of this study. I suggest that the author introduce it more systematically.

         (3) Theoretical analysis suggests further strengthening. At present, although the author's selection of indicators is carried out on the basis of some literatures, the selection of these indicators does not seem to be in dialogue with classical theories. For example, Table 1 the author can have a dialogue with the existing three living theories (ecology, production and life).

         (4) The logic used in the author's approach suggests a separate paragraph in the study design section. What are the key scientific questions to be addressed? What methods are needed to solve these problems, and what is the logical order among them? This is more conducive to readers to understand the design of the whole article.

         (5) It is suggested that the last part should be changed to the research conclusion and policy enlightenment. It is suggested that the author combine the research results to condense some policy implications.

Author Response

First of all, thank you very much for putting forward valuable modification opinions to this paper, which makes the author think more deeply about the shortcomings of the manuscript. According to your comments, the author modified and improved one by one. The specific modifications are as follows:

Point 1: The marginal contribution of research needs to be further clarified. It is suggested that the last paragraph of the introduction should systematically explain the marginal contribution of this research, for example, from the perspective of research, research content, research methods and other aspects to do a systematic discussion.

Response 1: The following text was added at the end of the introduction to illustrate the marginal contribution of this research: “From the perspective of research, this paper takes the agricultural transformation of the typical metropolis, Guangzhou, as a case to study its continuous carbon effects, and explains it according to the practicalities of urban agricultural development in Guangzhou, which broadens the research perspective in terms of space, time and effect. In terms of research content, this paper emphatically notes that the multifunctional transformation of urban agriculture can be divided into different stages, where the production, economic, social and ecological functions played by urban agriculture were different, and different carbon effects were generated. This enriches the exploratory research on the ecological effects of agricultural regional systems and makes contributions to the search for carbon emission reduction strategies in metropolitan areas. In terms of research methods, this paper applies the Granger causality analysis method and the grey association model to explore the long-term time series interaction relationship between urban agricultural multifunctionality and carbon effects, which is suitable and innovative.”

Point 2: Why we should focus on the multifunctionality of urban agricultural may need to say more. Compared with the well-known contribution of traditional rural agricultural production and urban industrial production to carbon emissions, the functionality of urban agricultural multifunctionality that the author focuses on is special. That is the highlight of this study. I suggest that the author introduce it more systematically.

Response 2: In the introduction of urban agriculture, the following text was added to illustrate its particularity: “Compared with traditional agricultural production, urban agriculture has stronger scientific and technological advantages and more abundant sales channels, so it has higher production efficiency and economic benefits. The service industry and its complete industrial chain derived from it further generate benign social effects. The transformation of production modes, the application of ecological economy and the deepening of the low-carbon concept have made urban agriculture generate carbon effects different from traditional agriculture. On the other hand, urban industrial production only pursues economic benefits, burns a large amount of fossil fuels and emits carbon dioxide, which is a socio-economic system completely controlled by human activities, and therefore it is difficult to make contributions to carbon sequestration. In contrast, urban agriculture is a regional system where human beings and nature coexist in harmony and supplement each other. While pursuing production benefits through science and technology, it also takes the positive impact of crops on the ecological environment into account, and therefore promotes the carbon balance of the ecosystem.”

Point 3: Theoretical analysis suggests further strengthening. At present, although the author's selection of indicators is carried out on the basis of some literatures, the selection of these indicators does not seem to be in dialogue with classical theories. For example, Table 1 the author can have a dialogue with the existing three living theories (ecology, production and life).

Response 3: The following text was added to illustrate the dialogue between the “three living” theory and the construction of the index system: ”The development goal of the regional land “production-living-ecological” space is "intensive and efficient production space, habitable and appropriate living space, and ecological space with pure and natural beauty". This conceptual framework covers biophysical processes, direct and indirect production, as well as spiritual, cultural, leisure and aesthetic needs [34]. Based on the ideological connotation of the "three living" space, the index system of urban agricultural functions, namely production, economy, society and ecology, was constructed.”

Point 4: The logic used in the author's approach suggests a separate paragraph in the study design section. What are the key scientific questions to be addressed? What methods are needed to solve these problems, and what is the logical order among them? This is more conducive to readers to understand the design of the whole article.

Response 4: The following text was added to illustrate the key scientific questions and corresponding methods: ”This paper aims to address the following scientific questions. (1) What kind of multifunctional transformation process did urban agriculture experience in Guangzhou? (2) What kind of carbon effects did urban agriculture generate in Guangzhou? (3) Is there an interaction between urban agricultural multifunctionality and carbon effects? How strong is the impact intensity? (4) What were the temporal characteristics of the interaction between urban agricultural multifunctionality and carbon effects? This paper follows the research logic of "process-effect-mechanism" and makes the following research design (Figure 2). First, agricultural transformation process was displayed by constructing evaluation index system of urban agricultural multifunctionality. At the same time, the continuous change process of emissions and sequestration generated by carbon sources and carbon sinks was examined by calculation. Then, the Granger causality analysis method was applied to test the interaction between the two, and an impulse response function was used to examine the impact intensity. Next, the grey association model was used to reveal the temporal characteristics of the interaction between the two. Finally, carbon effects generated by the multifunctional transformation of urban agriculture was explained according to the practicalities of urban agricultural development in Guangzhou on the basis of the literature.”

Point 5: It is suggested that the last part should be changed to the research conclusion and policy enlightenment. It is suggested that the author combine the research results to condense some policy implications.

Response 5: The following policy implications was added at the end of the paper: “Compared with traditional rural agriculture and urban industrial production, urban agriculture has made positive contributions to carbon emission reduction and carbon sequestration increase due to its own particularity. According to the emission reduction and sequestration increase effect generated in the process of the multifunctional transformation of urban agriculture, the policy implications on how to solve the problem of excessive carbon dioxide emissions through agriculture in metropolitan areas are put forward as follows. First, by increasing efforts in technological research and development, reduce carbon emissions in the production process of soil tillage, irrigation, fertilization, spraying and covering agricultural film, and explore the optimal production mode of carbon sink. Second, innovate the operation mechanism of agricultural enterprises, realize the integrated development of the three industries, build the cooperation platform of all stakeholders, make farmers benefit from the development of agriculture and the integration of industry, and provide technical training and demonstration consulting for farmers regularly, which is conducive to transforming their production and life concepts to low-carbon. Third, deepen the reform of agricultural modernization, introduce modern agricultural projects such as digital economy and smart agriculture, open up internet sales channels, and guide the operators to broaden the industrial chain and create brand agricultural products under the logic of pursuing the maximum profit, so as to promote their research on the carbon neutral green production mode of brand agricultural products.”

Reviewer 2 Report

The topic is in accordance with the Journal’s themes. The aim of the research is extremely clear, as well as the literature review. However, despite its undoubted interest as a relevant topic, too often the text results as redundant and sometimes repetitive, it is suggested to synthesize it and eliminate redundancies and repetitions to rise the clearness of reading. The objective of the study is partially achieved. It deserves to be clarified more in the expected output, which is few developed despite the clarity of the preliminary studies and analyses. The literature review is quite adequate, but please check if all of the references are quoted in the text. The applied methodology is appropriate but not well stated and explained in the conclusions of the article. It is suggested to manage better the frame of the methodology and maybe to list (instead to treat them as texts) some of the outputs/topics to ease the reading. The findings are interpreted but little expressed. Both language and structure are quite clear, but a consistent review of the English language and form is requested.

Overall, the paper topic is extremely relevant, and the adopted methodology organized, it deserves to be published but it needs of consistent reviews and re-reading to avoid messes and unclarity, mostly, in the explication of the methodology, the case study and the final output of the research work.

Definitely, they must be improved and better described (with a clear order and framework). Some of the assumptions are too generic and only a study-area related. Furthermore, some of the treated topics are just drafted and not addressed in their substantial meanings. Please, see the above suggestions and comments split within the review too

Author Response

First of all, thank you very much for putting forward valuable modification opinions to this paper, which makes the author think more deeply about the shortcomings of the manuscript. According to your comments, the author modified and improved one by one. The specific modifications are as follows:

Point 1: The literature review is quite adequate, but please check if all of the references are quoted in the text. Both language and structure are quite clear, but a consistent review of the English language and form is requested.

Response 1: Some references were added. All of the references were checked to ensure them correctly quoted in the text. This paper was read repeatedly to ensure the correct English language and form. The expressions was revised to be consistent.

Point 2: The objective of the study is partially achieved. It deserves to be clarified more in the expected output, which is few developed despite the clarity of the preliminary studies and analyses. The applied methodology is appropriate but not well stated and explained in the conclusions of the article. The findings are interpreted but little expressed.

Response 2: The third conclusion was revised: “According to the urban agricultural multifunctionality in Guangzhou and the change of its associative degree with carbon effects from 2002 to 2020, the development process of their interaction can be divided into three stages: The first stage was from 2002 to 2006, when agriculture in Guangzhou was dominated by the production function, and the economic and social functions were relatively weak. In this stage, the associative degree between agricultural multifunctionality and carbon sequestration was at a low level. The associative degree between economic function and carbon emissions was high and increasing. The associative degree between other functions and carbon emissions was low. The second stage was from 2007 to 2015. The production function declined steadily, and the economic and social functions gradually increased to the same level as the production function. In this stage, the associative degree of the production function with both carbon sequestration and carbon emissions was at a higher level. The associative degree of the economic function with both carbon sequestration and carbon emissions was declining. The associative degree between other functions and carbon emissions was low. The third stage was from 2016 to 2020. The production function further declined, and the economic and social functions exceeded the production function. Urban agricultural multifunctionality fully manifested. During this period, the associative degree of agricultural production function with both carbon sequestration and carbon emissions decreased slightly and stably, while the associative degree between other functions and carbon effects was low.”

Point 3: It is suggested to manage better the frame of the methodology and maybe to list (instead to treat them as texts) some of the outputs/topics to ease the reading.

Response 3: The following text was added to illustrate the key scientific questions and corresponding methods, and a framework of the methodology was designed: ”This paper mainly addresses the following scientific questions. (1) What kind of multifunctional transformation process did urban agriculture experience in Guangzhou? (2) What kind of carbon effects did urban agriculture generate in Guangzhou? (3) Is there an interaction between urban agricultural multifunctionality and carbon effects? How strong is the impact intensity? (4) What are the temporal characteristics of the interaction between urban agricultural multifunctionality and carbon effects? This paper follows the research logic of "process-effect-mechanism" and makes the following research design (Figure 2). First, agricultural transformation process was displayed by constructing evaluation index system of urban agricultural multifunctionality. At the same time, the continuous change process of emissions and sequestration generated by carbon sources and carbon sinks was examined by calculation. The Granger causality analysis method was applied to test the interaction between the two, and an impulse response function was used to examine the impact intensity. The grey association model was used to reveal the temporal characteristics of the interaction between the two. Finally, the carbon effect generated by the multifunctional transformation of urban agriculture was explained according to the practicalities of urban agricultural development in Guangzhou on the basis of the literature.”

Point 4: Despite its undoubted interest as a relevant topic, too often the text results as redundant and sometimes repetitive, it is suggested to synthesize it and eliminate redundancies and repetitions to rise the clearness of reading. Some of the assumptions are too generic and only a study-area related. Furthermore, some of the treated topics are just drafted and not addressed in their substantial meanings. Please, see the above suggestions and comments split within the review too.

Response 4: The whole text was carefully combed to ensure the clearness of reading and the logical smoothness of illustration of key scientific questions.

Reviewer 3 Report

The paper is generally good and interesting for the reader. Here are some detail remarks:

-       Table 1 – How weights were established?

-       Formula (6) is not clear. Especially, how min and max operators are used? What is rho? Is it the original formula or taken from somewhere (reference?)? I think some written explanation is needed.

-       Chinese characters are visible on some figures, when the cursor is placed on the graph

-       Table 3 and some next ones. “Test type (N,T,K)” would be better in the title row (if it is true)

-       Table 5 – Do not be afraid of 0.0000 notation for p value. It’s much better than say 2*10-8

Author Response

First of all, thank you very much for putting forward valuable modification opinions to this paper, which makes the author think more deeply about the shortcomings of the manuscript. According to your comments, the author modified and improved one by one. The specific modifications are as follows:

Point 1: Table 1—How weights were established? 

Response 1: The following text was added to the end of 2.3.1: “The weight of each index was obtained by the entropy weight method.”

Point 2: Formula (6) is not clear. Especially, how min and max operators are used? What is rho? Is it the original formula or taken from somewhere (reference?)? I think some written explanation is needed.

Response 2: This method draws on the existing literature [43-45] and the formula was checked to ensure its correctness. The usage of min and max operators is generally accepted. ρ is the discrimination coefficient, with the value in (0, 1). The smaller ρ is, the greater difference between the correlation coefficients is, and the stronger the discrimination ability is. Generally take ρ = 0.5. The introduction of ρ was added in the note of Formula (6).

Point 3: Chinese characters are visible on some figures, when the cursor is placed on the graph.

Response 3: All the graphs were checked to ensure their correctness.

Point 4: Table 3 and some next ones. “Test type (C,T,K)” would be better in the title row (if it is true)

Response 4: This suggestion was adopted and (C,T,K) was added in the title row.

Point 5: Table 5 – Do not be afraid of 0.0000 notation for p value. It’s much better than say 2*10-8

Response 5: This suggestion was adopted and 0.0000 was used in Table 5.
